# The Proposed Neurotoxin β-*N*-Methylamino-l-Alanine (BMAA) Is Taken up through Amino-Acid Transport Systems in the Cyanobacterium *Anabaena* PCC 7120

**DOI:** 10.3390/toxins12080518

**Published:** 2020-08-13

**Authors:** Zi-Qian Wang, Suqin Wang, Ju-Yuan Zhang, Gui-Ming Lin, Nanqin Gan, Lirong Song, Xiaoli Zeng, Cheng-Cai Zhang

**Affiliations:** 1State Key Laboratory of Freshwater Ecology and Biotechnology and Key Laboratory of Algal Biology, Institute of Hydrobiology, Chinese Academy of Sciences, Wuhan 430072, China; wangziqian@ihb.ac.cn (Z.-Q.W.); suqinwanghn@outlook.com (S.W.); zhangjuyuan@ihb.ac.cn (J.-Y.Z.); linguiming@ihb.ac.cn (G.-M.L.); gannq@ihb.ac.cn (N.G.); lrsong@ihb.ac.cn (L.S.); 2University of Chinese Academy of Sciences, Beijing 100049, China; 3Institute AMU-WUT, Aix-Marseille Université and Wuhan University of Technology, Wuhan 430072, China; 4Innovation Academy for Seed Design, Chinese Academy of Sciences, Beijing 100049, China

**Keywords:** neurodegenerative disease, cyanobacteria, amino acids transport, BMAA resistance

## Abstract

Produced by cyanobacteria and some plants, BMAA is considered as an important environmental factor in the occurrence of some neurodegenerative diseases. Neither the underlying mechanism of its toxicity, nor its biosynthetic or metabolic pathway in cyanobacteria is understood. Interestingly, BMAA is found to be toxic to some cyanobacteria, making it possible to dissect the mechanism of BMAA metabolism by genetic approaches using these organisms. In this study, we used the cyanobacterium *Anabaena* PCC 7120 to isolate BMAA-resistant mutants. Following genomic sequencing, several mutations were mapped to two genes involved in amino acids transport, suggesting that BMAA was taken up through amino acid transporters. This conclusion was supported by the protective effect of several amino acids against BMAA toxicity. Furthermore, targeted inactivation of genes encoding different amino acid transport pathways conferred various levels of resistance to BMAA. One mutant inactivating all three major amino acid transport systems could no longer take up BMAA and gained full resistance to BMAA toxicity. Therefore, BMAA is a substrate of amino acid transporters, and cyanobacteria are interesting models for genetic analysis of BMAA transport and metabolism.

## 1. Introduction

The neurotoxin β-*N*-methylamino-l-alanine (BMAA) is a non-protein amino acid, first identified from *Cycas* in 1967 [1]. Although the underlying mechanism is not understood, the accumulation of BMAA in brain tissues is found to be associated with neurodegenerative diseases such as amyotrophic lateral sclerosis, Parkinson’s disease, and Alzheimer’s disease [2,3,4,5]. It is also postulated as one of the environmental factors that contribute to the relatively high rate of dementia in Finland [6]. BMAA is also widely detected in different cyanobacterial species including free-living and symbiotic strains, suggesting that BMAA is an important metabolite in these organisms [7], although some other reports failed to detect the presence of BMAA in cyanobacteria [8,9,10,11,12,13]. Cyanobacteria are widespread among different environmental habitats, including those in symbiosis with plants, and some cyanobacteria species are notorious for their ability to form dense blooms in eutrophic water bodies [14,15]. In addition, BMAA is also reported to accumulate through the food chain [16]. Therefore, the production of BMAA by cyanobacteria is considered as an important environmental factor affecting human health [17]. However, despite the extensive studies, the biosynthesis and metabolic pathways of BMAA remain largely unknown, and the hypothetical pathways recently proposed still lack experimental support [18,19].

The production of BMAA in cyanobacteria can be enhanced by environmental factors such as nitrogen starvation [20,21]. Curiously, at least two cyanobacterial strains, namely *Synechocystis* PCC 6803 (hereafter *Synechocystis*) and *Anabaena* PCC 7120 (hereafter *Anabaena*), are sensitive to BMAA exogenously added to the culture medium [22,23]. Since both strains can be manipulated genetically [24,25], their sensitivity towards BMAA opens the door to genetic studies of the biosynthesis and the metabolism of this neurotoxin. These findings also indicate that BMAA can enter cyanobacterial cells either passively or through an uptake system that remains to be identified. In general, amino acid uptake systems are well studied in cyanobacteria, especially in *Anabaena* [26,27]. This organism can use either a combined source such as ammonium and nitrate as a nitrogen source, or atmospheric N_2_ through nitrogen fixation confined in specialized cells called heterocysts. Heterocysts are formed upon deprivation of the combined nitrogen source in the growth medium [28]. Irrespective of the growth mode, *Anabaena* can take up various amino acids relying on ABC-type membrane transport systems [26,27]. Four amino acid transporters, with relatively weak substrate specificity, have been characterized in *Anabaena*: N-I, N-II, Bgt, and N-III (Appendix A). The N-I system is composed of five polypeptides, NatA (All1046), NatB (Alr1834), NatC (All1047), NatD (All1248) and NatE (All2912), and it is the major transport system for hydrophobic amino acids, although it can take up most of the 20 amino acids [29]. The N-II system is composed of NatF (Alr4164), NatG (Alr4165), NatH (Alr4166) and BgtA (Alr4167), and mainly recognizes acidic amino acids (Asp and Glu) and neutral polar amino acids. The Bgt system requires BgtA and BgtB (Alr3187), and recognizes mainly basic amino acids such as Lys and Arg. Finally, the N-III system encoded by the *alr2535*-*alr2541* gene cluster is mostly involved in the uptake of Gly. Although each system has preferred substrates, they have extensive overlapping functions. For example, all four transporters can recognize Gln to various extents [26]. N-II and Bgt systems also share BgtA as the common ATPase component [27]. N-III plays a minor role in amino acids uptake, since its inactivation affected the uptake rate of various amino acids only weakly, while N-I, N-II and Bgt together are responsible for more than 98% of the amino acid transport activity in *Anabaena*. Some of the amino acid transport components are also necessary for diazotrophic growth [26,27].

In this study, we used *Anabaena* as a model to determine the transport systems allowing BMAA to enter the cells by characterizing spontaneous mutations or creating specific mutants. Our results indicate that BMAA enters *Anabaena* cells through the proteinogenic-amino acid transport systems. A mutant with inactivation of three amino acid transport systems in *Anabaena* confers full resistance to BMAA while losing BMAA uptake capacity. This study paves the way for further understanding on the metabolism of BMAA in cells.

## 2. Results

### 2.1. Anabaena Is a Suitable Model for Genetic Studies of BMAA Metabolism

Previously, two independent reports described the inhibitory effect of BMAA on *Synechocystis* and *Anabaena*, respectively [22,23]. In order to select the most suitable report for further genetic characterization, we compared the growth of these two strains in the presence of different concentrations of BMAA in the growth medium. When grown with 25 μM or 50 μM of BMAA in BG11, *Synechocystis* displayed a much slower growth rate than the control with no BMAA added. A concentration of BMAA above 200 μM was needed to suppress completely the growth ability of *Synechocystis* (Figure 1A). By comparison, a concentration of BMAA as low as 5 μM could already have an inhibitory effect on the growth of *Anabaena* in BG11, and 25 μM of BMAA was enough to inhibit completely its growth (Figure 1B). *Anabaena* was even more sensitive to BMAA when it was grown in BG11_0_ medium in which no combined nitrogen was present. Indeed, BMAA as low as 6 μM was enough to block completely the growth ability of this strain (Figure 1C). Our results are consistent with previous publications [22,23], and *Anabaena*, highly sensitive to BMAA, appeared to be more suitable for further genetic studies than *Synechocyctis*.

### 2.2. Isolation of BMAA Resistant Mutants

In order to gain insight into the mechanism of BMAA toxicity and metabolism, we selected BMAA resistant (BMAA^r^) mutants using *Anabaena*. Compared with liquid cultures, we found that a higher concentration of BMAA was necessary to completely inhibit the growth of *Anabaena* on solid plates, especially when a high density of cells was plated. Therefore, a concentration of 100 μM of BMAA was used for the selection of BMAA^r^ mutants on solid media. To maximize the chance of obtaining independent mutants, 10 pre-cultures in a small volume were grown independently (see section on Method and Materials for more details). Finally, we obtained 16 mutants (M1-M16) resistant to BMAA, with each initial independent culture giving two–five clones.

The resistance of all 16 mutants to BMAA was compared under BG11 and BG11_0_ conditions in liquid media. As shown in Figure 2, in BG11 without BMAA, all mutants grew similarly as compared to the wild type except M13 which grew slightly more slowly. When 50 μM BMAA was supplied to BG11, all the mutants were able to grow while the wild type was unable to do so (Figure 2A). In BG11_0_ medium with or without BMAA, the WT strain showed the same growth pattern as in Figure 1C; however, a disparity in the growth rate was revealed for the 16 mutants. In the first category, several mutants were severely affected, either unable to grow (M12, M13) or growing badly (M1, M4, M16) even without BMAA. Upon addition of BMAA, none of the five mutants could grow. For the second category, four mutants (M2, M5, M7, and M14) could grow in BG11_0_ but resisted BMAA poorly. In the last category, the mutants (M3, M6, M8, M9, M10, M11, and M15) grew similarly in BG11_0_, no matter whether BMAA was present or not (Figure 2B). Consistent with the fact that *Anabaena* is more sensitive to BMAA in BG11_0_ than in BG11 (Figure 1C), characterization of different mutants also indicates that nitrogen regime strongly affected resistance levels to BMAA. For the mutants unable to grow in BG11_0_, either with or without BMAA, we also checked their ability to differentiate heterocysts. Heterocysts were observed in these mutants similarly to the wild type within 24 h after the deprivation of combined nitrogen (data not shown). As discussed later, the difference in BMAA resistance observed for some mutants was related to the requirement of amino acid transport systems for diazotrophic growth [27].

### 2.3. Spontaneous Mutations Conferring Resistance to BMAA in Anabaena Were Mapped on Genes Involved in Amino Acid Transport

To understand the mechanism of BMAA resistance, we chose four mutants (M1–4) for whole genome sequencing. These mutants were obtained from four different pre-cultures and displayed different resistance levels to BMAA (Figure 2B). A wild-type sample, saved before the isolation of mutants, was used as a control. The sequencing reactions had coverage between 97–99% for each strain. M4 displayed no difference to the control strain, suggesting that the mutation could have occurred in the parts of the genome uncovered by sequencing, or that the mutation was not found because of an unknown technical problem. For the remaining three strains, however, mutations were localized at *alr4147*/*bgtA* and *all1284*/*natD*, as compared to the control strain (Figure 3; Appendix A), and thus these mutations affected amino acid transport. M1 harbored one deletion mutation in the coding region of *bgtA* at position A354, producing thus a frameshift mutation, and an additional transition mutation (change from A to G) at position 716 in the coding region of *natD*, leading to the replacement of a Trp residue by a Cys residue. M2 accumulated two mutations, one a C→T transition at position 253 of *bgtA*, producing a stop codon instead of an Arg residue, and the second also corresponding to a C→T transition close to the end of the *natD* coding region. In the mutant M3, a frameshift mutation due to the deletion of a single base at position 652 of the coding region was detected in *natD*. Thus, all three mutants affected two genes, both of which encoded components of amino acid transport systems in *Anabaena*.

Since the three mutants affected two amino acid transport systems, we thus examined the two corresponding genes by PCR and sequencing in the remaining 12 mutants isolated at the same time (Figure 3; Appendix A). Three, namely M6, M7 and M14, had no mutations at either *bgtA* or *natD*, thus the genetic basis of their resistance needed to be further analyzed. For the others (M5, M8–13, M15, and M16), each had one or two mutations mapped at either *bgtA* or *natD*. M5 had the same mutation as in M1 on *natD*. M8 had a T→C transition at a position close to the 5′ coding region of *bgtA*. M9 accumulated two mutations, one the same as in M2 and M10 in *bgtA*, and the other one with three bases inserted into *natD*. M11, M12 and M15 all had a mutation that was the same as in M3 in *natD*. M16 had a frameshift mutation due to the deletion of a single base at position 649 of the coding region in *natD*. Finally, M13 had an additional mutation in *bgtA* corresponding to an insertion of two bases (Figure 3; Appendix A). The occurrence of mutations at the same genic locus in multiple independent mutants strongly suggested that *bgtA* and *natD*, and the corresponding amino acid transport systems, were involved in BMAA resistance.

### 2.4. Exogenously Supplied Amino Acids in the Growth Medium Relieve the Toxic Effect of BMAA in Anabaena

We further used two strategies to confirm the relationship between amino acid transport and BMAA resistance. The first was a competition assay between BMAA and amino acids, and the second was the targeted mutation of amino acid transporter genes and their characterization in terms of BMAA toxicity.

The mutations detected in two genes involved in amino acid transport in nine BMAA^r^ mutants strongly suggested that the N-I system involving NatD, as well as the NII and the Bgt systems that share BgtA, participated in BMAA transport. If this hypothesis is true, standard amino acids would provide a protective effect on the cells incubated with BMAA. The four described amino acid transport systems in *Anabaena* have overlapping substrate specificity [26], and we chose five amino acids with different biochemical characteristics for the competition experiments with BMAA: Lys (basic amino acid); Asp (acidic amino acid); Ser and Pro (neutral polar amino acid), and Gly (hydrophobic amino acid). Asp is taken up by NII; Lys by Bgt; Ser by N-I and N-II; Gly by N-I, NII and N-III; and Pro by N-I and N-III (Appendix A). We first examined the protective effect of each amino acid with a ratio relative to BMAA at 10, with 25 μM of BMAA in BG11 and 10 μM in BG11_0_, because *Anabaena* is much more sensitive to BMAA in BG11_0_ than in BG11 medium (Figure 1). As shown in Figure 4A,B, *Anabaena* grew similarly when incubated with or without each of the five amino acids in both BG11 and BG11_0_. In the presence of BMAA alone, *Anabaena* failed to grow, as expected, in the two-culture media. However, when each of the five amino acids was added, it was able to grow except when Lys was supplied in BG11_0_. Thus, all five amino acids protected *Anabaena* from BMAA toxicity in BG11, with Asp having the weakest protective effect. Lys is taken up by the Bgt system [27], and the lack of protection by Lys under diazotrophic conditions (BG11_0_) suggested that the Bgt system was less involved in BMAA uptake under such conditions.

To better evaluate the protective effect of each amino acid, we performed the competition experiment by varying the ratio between BMAA and each amino acid (Figure 4C,D). In BG11, as the concentration of the amino acids increased, the toxicity of BMAA decreased, with Lys, Ser and Gly showing the best effect. Asp was the worst protector under such a condition, as seen in Figure 4A. In BG11_0_, although the protective effect was less evident, high concentrations of each amino acid protected the growth of *Anabaena* in the presence of BMAA, only Lys displaying almost no effect. Thus, almost all tested amino acids could compete with BMAA, further indicating that BMAA enters the cells of *Anabaena* through the amino acid transport system.

### 2.5. Inactivation of the Three Major Amino Acid Transport Tystems Conferred Full Resistance to BMAA

Although we obtained several BMAA^r^ mutants with spontaneous mutations, these mutants affected two of the four amino acid transport systems in *Anabaena*. Therefore, to provide further genetic evidence on the involvement of amino acid transport systems in BMAA uptake and toxicity, we constructed several mutants inactivating each of the four amino acid transport systems in *Anabaena.* To inactivate *natA*, *bgtA* and *bgtB,* we replaced a large part of the coding region by an antibiotic resistance marker, and deleted the coding region of *natD*, *natG* and *natI* by using the recently available Cpf1-based markless gene editing technique [30,31] (Appendix A; Appendix A). We also created a double mutant in which both *natA* and *bgtA* were inactivated, thus producing a mutant defective in the three major amino acid transport systems in *Anabaena*, namely N-I, N-II and Bgt [27]. All mutants were verified by PCR for the complete segregation (Appendix A).

The resistance of all seven mutants to different concentrations of BMAA was first compared in BG11 (Figure 5). While the wild type started to display a slower growth in the presence of 10 μM of BMAA and an arrest of cell growth above 25 μM, as in Figure 1, four single mutants (*ΔnatA*, *ΔnatD*, *ΔnatG* and *ΔbgtA*) could resist up to 25 μM of BMAA, and even grew, albeit slowly, at 50 μM of BMAA. The *ΔbgtB* mutant showed almost no resistance to BMAA, and *ΔnatI* could resist only up to 25 μM of BMAA. These results indicated that in BG11 medium, N-I and N-II were the two most important systems involved in BMAA transport. This conclusion was further supported by the phenotype of the double mutant *ΔnatAΔbgtA*, in which N-I, N-II and Bgt systems were all inactivated. The double mutant was fully resistant to BMAA since a concentration of BMAA as high as 100 μM had little effect on its growth.

We also tested the resistance of these mutants to BMAA in BG11_0_ (Figure 5). It is known that some of the amino acid transporter genes are involved in diazotrophic growth, especially those belonging to N-I and N-II [26,27]. Thus, some mutants displayed an already weaker growth in BG11_0_ compared to the wild type (*ΔnatA*, *ΔnatD*), even hardly growing at all (*ΔnatG*, *ΔnatI*). Each single mutant showed a low-level or undetected level of resistance to BMAA. However, the double mutant *ΔnatAΔbgtA* was able to sustain a slow growth even up to 100 μM of BMAA, comparable to the mutant grown without BMAA. Therefore, the inactivation of the three major amino acid transport systems also provided resistance to BMAA under diazotrophic conditions.

### 2.6. BMAA Uptake Assay in the Wild Type and the Mutants

To examine the extent of BMAA uptake through the three major amino acid transport systems, we directly measured the uptake of BMAA in wild type and *ΔnatAΔbgtA* mutant strains by ultra-high performance liquid chromatography with tandem mass spectrometry detection (UPLC-MS/MS) coupled with derivatization using 6-aminoquinolyl-N-hydroxysuccinimidyl carbamate (AQC) [32]. Under our assay conditions with the UPLC-MS/MS technique, the limit of detection (LOD) of BMAA standard is evaluated to be 0.017 pmol per injection (or 0.5 μg/L) and the limit of quantification (LOQ) is at 0.042 pmol per injection (or 1.25 μg/L) (Figure 6A). The LOD determined in our experiments is among the lowest when compared to other reported results which range from 0.016 pmol to 65 μmol per injection, except for one laboratory which reported a much lower LOD at 0.01 pg (8.5×10^−5^ pmol) per injection [33,34].

We then applied this technique to the detection of BMAA uptake in *Anabaena* and *ΔnatAΔbgtA* double mutant cultures. After incubating the cells at exponential phase with 50 μM BMAA in BG11 medium for 0, 1, 7 and 15 min, the cell extracts were prepared and used for the detection of BMAA. The results showed that, as the exposure time increased, the amount of BMAA in cells of the wild type increased. However, for the mutant *ΔnatAΔbgtA*, no significant amount of BMAA could be found even after 15 min of incubation (Figure 6B,C). These results provided direct evidence that BMAA was taken up mainly through N-I and N-II systems in *Anabaena*.

## 3. Discussion

In this study, we provided evidence that BMAA enters the cells through the proteaginous amino acid transport systems in the cyanobacterium *Anabaena*. This conclusion was based on the sequencing of spontaneous mutants (Figure 2 and Figure 3), competition experiments between BMAA and various amino acids (Figure 4), as well as targeted inactivation of each of the four known amino acid transport systems (Figure 5). The double mutant *ΔnatAΔbgtA*, in which the three major transport systems accounting for 98% of the amino acids transport capacity were abolished, could not take up exogenous BMAA anymore, and had a full immunity against the toxic effect of BMAA (Figure 6).

So far, neither the biosynthetic nor the metabolic pathway of BMAA is known. *Anabaena*, with a high sensitivity to BMAA and the availability of a genetic system, constitutes an original and useful model to identify genetic and biochemical components involved in BMAA biosynthesis or metabolism. Spontaneous mutants resistant to BMAA arose frequently under our assay conditions (Figure 2 and Figure 3). The whole genome sequencing technique offered the possibility of identifying relatively easily the mutations responsible for the corresponding phenotypes. Our M1-M4 mutants have different levels of BMAA resistance (Figure 2), and the genome sequencing results of these four representative mutants indicated that they had different mutation sites (Appendix A). Although these mutations were mapped on two genes, different types of mutations, leading to different forms of corresponding proteins, were detected. For examples, M1 and M3 both have mutations on the gene *natD*, but M1 changed Tyr239 of NatD to Cys, while M3 expressed a truncated form of NatD. According to our experience, it was important to minimize the occurrence of genetic changes that may otherwise accumulate following successive sub-culturing. In order to obtain different mutants, the wild-type strain, cultured and saved just before the isolation of the mutants, was used as a control in genomic sequencing, and independent cultures were amplified for isolation of different clones (Figure 3). Indeed, the same *Anabaena* strain, maintained and cultured in different laboratories, tends to display differences in the genomic sequences, indicating that genetic changes occur constantly in this organism [35].

*Anabaena* cells are much more sensitive to BMAA under combined-nitrogen depleted conditions than under combined-nitrogen-replete conditions (Figure 1). The toxicity of BMAA under diazotrophic conditions has already been reported, and the same report showed that BMAA inhibited the nitrogenase activity [23]. It has been found that BMAA can influence the expression of some genes related to heterocyst development [36]. However, nitrogen fixation is unlikely to be the only target of BMAA since the inhibitory effect is also observed when *Anabaena* cells are grown in the presence of a combined-nitrogen source. Some of the amino acid transport systems are required for cell growth under diazotrophic conditions [26], and they are responsible for intercellular exchanges of nitrogen compounds in the form of amino acids, in particular between N_2_-fixing heterocysts and vegetative cells [37]. BMAA may interfere with the intercellular exchange of amino acids, thus making *Anabaena* filaments more sensitive to this compound under such conditions. Our genetic analysis indicates that, among the four transport systems, N-I and N-II are the two most important for BMAA uptake. When *bgtB* was inactivated alone, little resistance to BMAA was found (Figure 5). All single mutants related to either N-I or N-II could confer a certain level of resistance to BMAA, and the *bgtA* mutant affecting both the Bgt and the N-II systems had a comparable resistance to BMAA as that of the *natG* mutant affecting only N-II. Thus, the strong resistance of the double mutant *ΔnatAΔbgtA*, as shown in Figure 5, is derived mostly from the inactivation of the N-I and the N-II transport system.

Since the first report on the production of BMAA by cyanobacteria and its potential roles in the development of neurodegenerative diseases, considerable interest has developed in the scientific community on the identification of BMAA from cyanobacterial samples. The identification of BMAA has been reported in various species of cyanobacteria, including *Anabaena,* with a concentration determined at 32 μg/g DW for BMAA [7]. The UPLC-MS/MS method used in this study gave a detection limit among the lowest so far reported. However, under our conditions, no endogenous BMAA signal could be found in the wild type or *ΔnatAΔbgtA* double mutant strains, although BMAA was found in the wild type after exposure to exogenous BMAA (Figure 6). *Anabaena* may produce too little BMAA for detection, or the strain maintained under laboratory conditions has lost the ability to produce BMAA. Note also that some of the literature has also reported the failure to detect BMAA in cyanobacteria [8,9,10,11,12,13]. More rigorous experiments are needed to answer this question.

All the BMAA^r^ mutations identified either by direct genomic sequencing or sequencing of PCR fragments accumulated one or two mutations on two genes, *bgtA* and *natD*. Other genes known to be involved in amino acid transport, and also in BMAA resistance as shown here, have not yet been identified by looking for spontaneous mutations. Therefore, *bgtA* and *natD* appeared to be mutation hot spots under our assay conditions. Whole genome sequencing on mutants, such as M6, M7 and M14, for which no mutation has been found in either of these two genes (Figure 3), together with other mutants isolated by large-scale screening, could prove interesting, as they may reveal novel mechanisms involved in BMAA resistance, and potentially also in the control of amino acid metabolism. This could be achieved by additional screening of BMAA-resistant, or hyper-sensitive mutants, followed by characterization of the genetic factors involved. Such studies may also be important for the understanding of BMAA toxicity in humans, since it has been reported that amino acid transporters could be responsible for BMAA uptake in human cells, but genetic evidence is still lacking [38,39]. Amino acid transporters are highly conserved in eukaryotes and prokaryotes, the available genetic systems in cyanobacteria, and the information so obtained could therefore be useful for the understanding of the BMAA effects in other organisms too.

## 4. Materials and Methods

### 4.1. Strains and Growth Conditions

All the strains used in this study are described in Appendix A. *Anabaena* and its derivatives were grown in the mineral medium BG11 or BG11_0_ (same as BG11 but free of combined nitrogen), as described previously [40]. *Synechocystis* was also grown in BG11 as reported [41]. When necessary, neomycin (2.5 μg/mL), spectinomycin (5 μg/mL) or streptomycin (2.5 μg/mL) was added into the medium. For growth curve measurement, strains were incubated in liquid medium under artificial illumination with shaking at 180 rpm. The constant light intensity was 1800 Lux. For BMAA and amino acid competition experiments, liquid cultures in small plastic vials were incubated with illumination and shaking as indicated above, followed by imaging from the bottom of the vials.

### 4.2. BMAA, Other Chemicals and Molecular Biology Tools

L-BMAA hydrochloride was ordered from Wuhan Chuanliu Biotechnology Co., Ltd (Wuhan, China). The stock solution of BMAA (50 mM, dissolved in H_2_O) was preserved at −20 °C. AccQ-Tag Ultra Derivatization Kit for BMAA derivatization was from the Waters Corporation (Milford, MA, USA). All the amino acids used in this study were purchased from Sigma-Aldrich Co. LLC. (St. Louis, MO, USA). The restriction enzymes were from TaKaRa, and PCR-related products, ClonExpress II One Step Cloning Kit, and ClonExpress MultiS One Step Cloning Kit were all from the Vazyme Biotech Co., Ltd (Nanjing, China).

### 4.3. Screening of Spontaneous BMAA Resistance Mutants

Wild type *Anabaena* at logarithmic phase in liquid culture was used as starting material (parental strain). To maximize the chance of obtaining independent mutants, we inoculated the parental strain into 10 independent conical flasks containing 80 mL BG11 medium, with a dilution rate of 1:500. The remaining wild type *Anabaena* cells were saved and used to isolate DNA for genomic sequencing. When each culture was grown independently to optical density (OD) of around 1.5, the cells were plated separately onto BG11 agar plates containing 100 μM BMAA. All the single colonies grown on the plates were transferred to a new BMAA (100 μM)-containing agar plate, and grown-up colonies were the BMAA-resistant mutants. These isolated mutants were cultured in BG11 medium with 25 μM BMAA for further analysis

### 4.4. Construction of Plasmids, Mutants and Genomic Sequencing

Details are described at the section of Supplemental Experimental procedures in supporting information. The strains are listed in Appendix A, and all the oligonucleotides are presented in Appendix A. 

### 4.5. Sample Preparation for BMAA Detection

To test BMAA uptake, cells were cultured to exponential phase, then BMAA was added into the medium to a final concentration of 50 μM. Fractions of the culture, each of 20 mL, were withdrawn at time 0 (just before addition of BMAA), and 1 min, 7 min and 15 min, respectively, after the addition of BMAA. Cells were quickly harvested by filtration and washed in BMAA-free medium to remove residual BMAA. The sample preparation was performed according to Faassen et al. [42] with modifications. The harvested cells were boiled in 500 μL double distilled water for 20 min, and then broken through crusher FastPrep-24TM5G (6 m/s for 2 min). The cell lysis efficiency was checked by microscopy. Once the cells were totally broken, samples were centrifuged at a speed of 13,000 rpm for 30 min and the precipitate was discarded. The supernatant was mixed with HCl (final concentration of 20 mM) and centrifuged again. The volumes of the supernatant obtained from different samples were approximately 350 ± 20 μL. All samples were stored at −80 °C before further analysis.

### 4.6. BMAA Derivatization

All the samples were derived by AccQ-Tag Ultra Derivatization Kit following the manufacturer’s instructions. Briefly, 6-aminoquinolyl-N-hydroxysuccinimidyl carbamate (AQC) was firstly dissolved in 1 mL acetonitrile to prepare the AQC reagent solution. For AQC derivatization, each extracted sample (20 μL) described above or standard BMAA solution (20 μL) was mixed with 60 μL borate buffers in a clean tube. An aliquot (20 μL) of AQC reagent solution was added and mixed by vortex. After mixing, the samples were kept at room temperature for 2 min, and then heated at 55 °C for 10 min. The derivate samples were stored at 4 °C for further detection.

### 4.7. UPLC-MS/MS

Quantitative measurements of BMAA were carried out by using UPLC-MS/MS on an ACQUITY UPLC H-class-Xevo TQ MS system, equipped with an ACQUITY UPLC BEH C18 column (2.1 × 50 mm, 1.7 μm) (Water Corporation, Milford, MA, USA). Separation was achieved by using gradient elution at 0.15 mL/min with solvent A (0.1% formic acid in water) and solvent B (0.1% formic acid in methyl alcohol): 0.0 min = 80% A; 4.0 min = 85% A; 4.1 min = 80% A; 5.8 min = 80% A. The Column temperature was 40 °C. The conditions for mass spectrometer were as follows: electrospray mode: ESI+; Capillary (kV): 3.0; desolvation temperature: 350 °C; nitrogen Gas Flow Desolvation (L/h): 650; argon collision gas flow (mL/min): 0.12; source temperature: 150 °C. To identify BMAA, MRM (Multiple Reaction Monitoring) was used to monitor the transitions: 459.10 > 119.10 (CE = 25), 459.10 > 171.10 (CE = 30), 459.10 > 258.10 (CE = 30), 459.10 > 289.10 (CE = 15). All samples in this study were analyzed with injections of 20 µL. The detection limit (LOD) and quantification limit (LOQ) of BMAA were determined as follows: a series of diluted standard BMAA solutions (0.25, 0.5, 1.25, 2.5, 5, and 12.5 μg/L) were detected by the procedure described above. The lowest concentration distinguishable from the background was defined as LOD, and the lowest concentration that still keeps a linear relationship on the calibration curve was defined as LOQ [42].

## Figures and Tables

**Figure 1 toxins-12-00518-f001:**
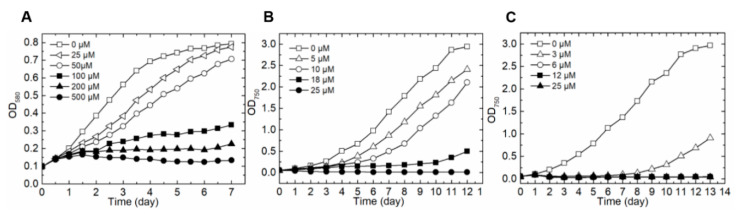
Effect of β-*N*-methylamino-l-alanine (BMAA) on the growth of the unicellular cyanobacterium *Synechocystis* PCC 6803 (**A**) and the filamentous diazotrophic strain *Anabaena* PCC 7120 (**B**,**C**). Growth curves were measured by following the optical density (OD) by spectrometer. The two strains were grown either in BG11 (**A**,**B**, containing nitrate as combined nitrogen source), or in BG11_0_ without combined nitrogen (**C**). The concentration of BMAA used for each culture was indicated.

**Figure 2 toxins-12-00518-f002:**
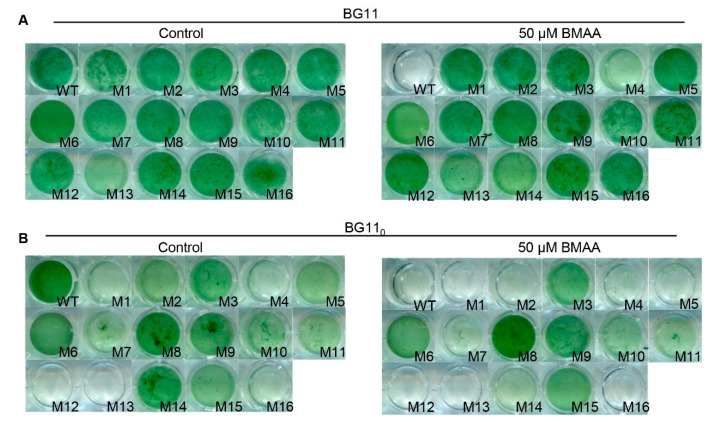
Characterization of spontaneous BMAA^r^ mutants isolated from *Anabaena*. The wild type (WT) or the mutants (from M1 to M16) were cultured in plastic vials in liquid medium with shaking, either in BG11 (**A**) or in BG11_0_ (**B**), without BMAA (control) or with 50 μM BMAA. All cultures were started with a similar optical density (OD) at 0.07 and after five days (in BG11) or seven days (in BG11_0_) of incubation; the vials containing the cultures were pictured.

**Figure 3 toxins-12-00518-f003:**
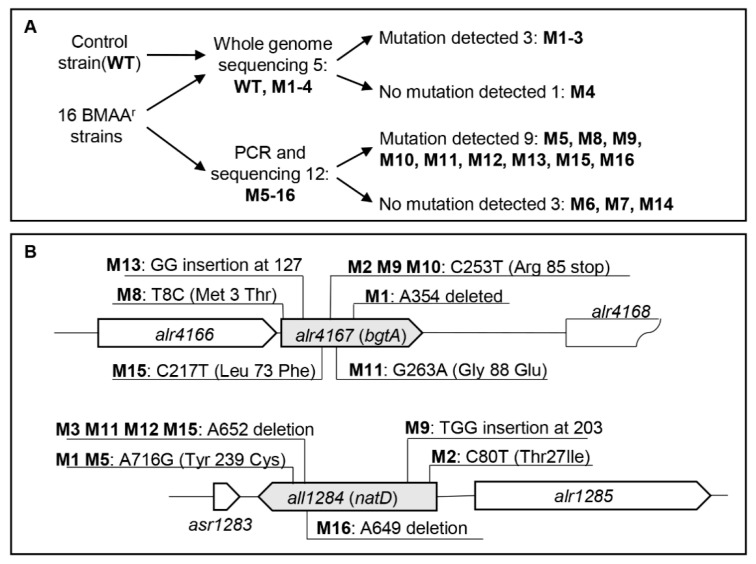
Determination of mutations in spontaneous BMAA^r^ mutants. (**A**) Schematic representation on the strategies used to determine the mutation site of the BMAA^r^ mutants. A total of 16 mutants were analyzed. The control strain and four mutants were characterized by whole genome sequencing, while the other mutants were checked by amplifying *bgtA* and *natD* by PCR, followed by sequencing. The number of mutants and results after sequencing are also indicated at each step. (**B**) Map of mutation sites in either *bgtA* or *natD*. The position of the mutation site, relative to the first base of each ORF (open-reading frame) is indicated. For transition mutations (example M8), the first letter is the base found in the control strain (WT), followed by the number indicating the position within the ORF, and the second letter indicates the base found in the mutant. The change in amino acid is also marked in the parenthesis. Some mutants have insertion (example M13), or deletion (example M1), and the relative position and the base affected are also marked.

**Figure 4 toxins-12-00518-f004:**
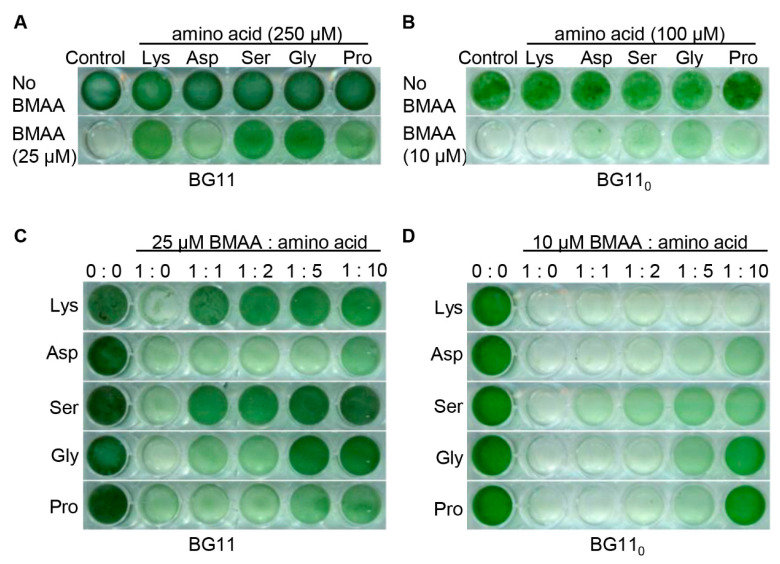
Protective effect of various amino acids against the toxic effect of BMAA. (**A**) Competition experiments with BMAA and amino acids in BG11. *Anabaena* was cultured in liquid medium with shaking in plastic vials, without BMAA (control) or with 25 μM BMAA. The concentration of each amino acid is 250 μM. After five days of incubation, the cultures were imaged. (**B**) Competition experiments with BMAA and amino acids in BG11_0_ deprived of combined nitrogen. The experiments were carried out similarly to (**A**) except the concentration of each amino acid is 100 μM. (**C**,**D**) Similar competition experiments as in A and B, except that the ration between BMAA and each amino acid varied from 0:0, to 1:0, 1:1, 1:2, 1:5, and 1:10. All cultures were started with a similar OD at 0.07 in BG11 (**C**) or BG11_0_ (**D**). After seven days of incubation, the cultures were imaged.

**Figure 5 toxins-12-00518-f005:**
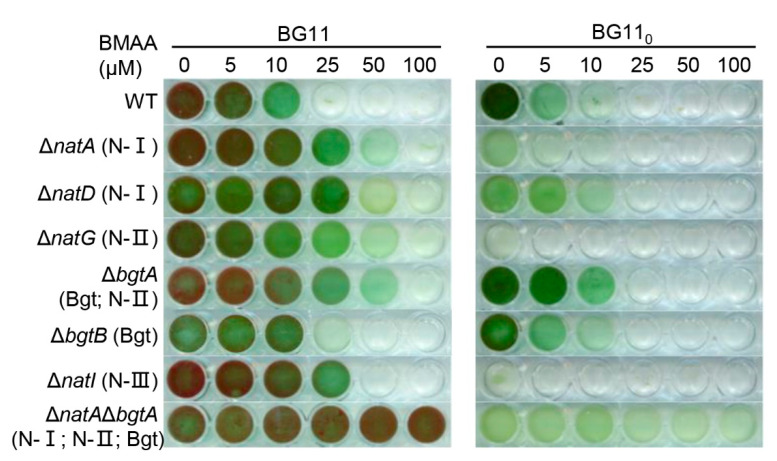
Effect of targeted mutation of genes encoding amino acid transport systems on the toxicity of BMAA. The wild type (WT) of *Anabaena* and seven mutants inactivating one or two genes involved in amino acid transport were incubated in BG11 or BG11_0_ with increasing concentration of BMAA. For each mutant, the corresponding affected transport system (N-I, N-II; N-III, or Bgt) is indicated in parenthesis. All cultures were started with a similar OD at 0.07, under illumination and shaking, and imaged after 12 days of incubation.

**Figure 6 toxins-12-00518-f006:**
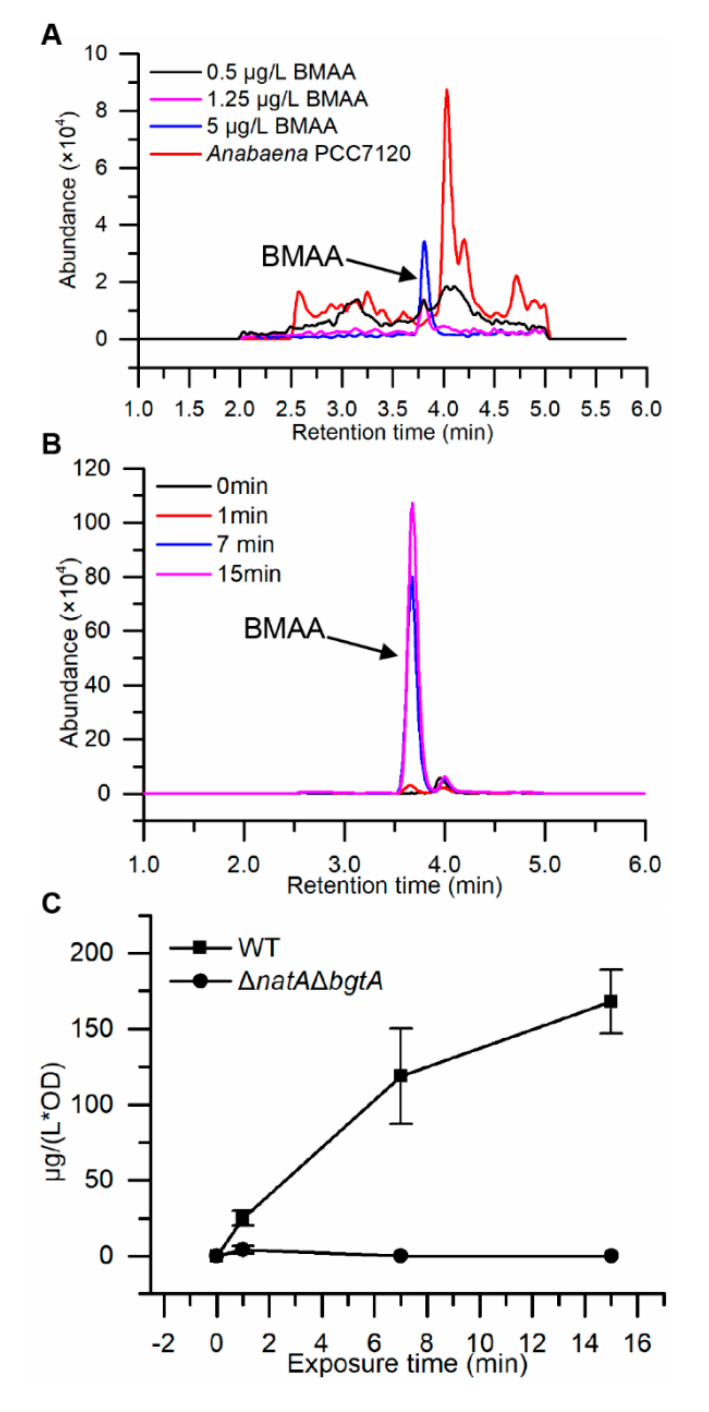
BMAA detection in *Anabaena* by ultra-high performance liquid chromatography with tandem mass spectrometry detection (UPLC-MS/MS) and the uptake of BMAA by *Anabaena*. (**A**) The limit of detection (LOD) and the limit of quantification (LOQ) of BMAA standard under our assay conditions. (**B**) Chromatograms for extracted samples of the wild-type *Anabaena* cells exposed to BMAA for different time points (presented with lines of different colors indicating exposure time 0, 1, 7, 15 min). (**C**) The uptake of BMAA by wide-type *Anabaena* and the mutant *ΔnatAΔbgtA*. The lines show the change of the amount of BMAA in WT *Anabaena* and *ΔnatAΔbgtA* with BMAA incubation time increasing. Error bars denote standard deviations for triplicates.

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
