# Peer review of "The Proposed Neurotoxin β-N-Methylamino-l-Alanine (BMAA) Is Taken up through Amino-Acid Transport Systems in the Cyanobacterium Anabaena PCC 7120"

_toxins, 2020, doi:10.3390/toxins12080518_

Round 1
Reviewer 1 Report
This is a comprehensive and well-designed study examining the role of amino acid transporters in mediating the transport of BMAA and hence, BMAA toxicity in 2 strains of cyanobacteria –– Synechocystis PCC 6803 and Anabaena PCC 7120.
Orthogonal experiments confirm a role for amino acid transporters in the movement of BMAA across the cell of Anabaena. This is achieved by spontaneous mutations as well as the construction of several mutants that render critical amino acid transporters inactivate, abrogates BMAA toxicity, and prevents the accumulation of BMAA inside the cells.
This is an interesting and well-designed study that advances the literature, since still so little is known about the metabolic and synthetic pathways for BMAA in cyanobacteria. I have made some minor suggestions, but once these corrections are addressed, I recommend it for publication.
----------
Line 7: Abstract: “Neither the underlying mechanism of its neurotoxicity….understood.” > “underlying mechanism of its neurotoxicity” in humans? Then no, this is not correct, there is a lot of evidence for mechanisms of neurotoxicity in humans. If the authors mean cyanobacteria, then this is true. But please rewrite the sentence to clarify.
Line 27: Although the underlying mechanism is not understood, the accumulation of BMAA in brain tissues is found to be associated to neurodegenerative diseases such as amyotrophic lateral sclerosis, Parkinson’s disease, and Alzheimer’s disease [2, 3]. The authors should cite some more recent references here, for example; https://doi.org/10.1098/rspb.2015.2397 and 10.1371/journal.pone.0213346
Line 97: “Compared with liquid cultures, we found that a higher concentration of BMAA was necessary to completely inhibit the growth of Anabaena on solid plates, especially when high density of cells was plated. “< data not shown? How did you ascertain this? Please briefly describe.
Line 110; mutants were severely affected, either unable to grow (M12, M13) or growing badly (M1, M4, 16) < insert “M” before 16.
Line 130: “To understand the mechanism of BMAA resistance, we chose 4 mutants (M1-4) for whole-genome sequencing……..” Please provide the details of the whole genome sequencing. Details of this should be inserted into the supplementary data. Please describe the methodology.
Line 133: “The sequencing reactions had coverage between 97-99% for each strain.” Please provide a description of the methods.
In figure 2 and 4, the authors show growth inhibition by BMAA as a visual. Can the authors explain why they didn’t also measure the growth using OD, and graph these data, as per figure 1? This would complement the images well.
Line 304; “It has been found that BMAA can influence the express…” < change to expression
Reviewer 2 Report
The manuscript provides evidence that the uptake and toxic effects of BMAA on cyanobacteria is mediated by specific amino acid transporters. The studies are well designed, executed, and presented. There is on major area of concern.
The concern is regarding the importance of the studies. The authors need to provide a better explanation of the relevance of the current studies. If the studies are important because of the potential relevance of BMAA to neurodegenerative diseases in humans it should be described why that is true. Are the transporters in cyanobacteria the same as in humans? Do the studies provide evidence of the mechanism of BMAA toxicity? The authors indicate that the studies are important for studying the metabolism of BMAA. It should be more clearly indicated how this would be determined.
